# MANI-PURE: MAGNITUDE-ADAPTIVE NOISE INJECTION FOR ADVERSARIAL PURIFICATION

## ABSTRACT

Adversarial purification with diffusion models has emerged as a promising defense strategy, but existing methods typically rely on uniform noise injection, which indiscriminately perturbs all frequencies, corrupting semantic structures and undermining robustness. Our empirical study reveals that adversarial perturbations are not uniformly distributed: they are predominantly concentrated in high-frequency regions, with heterogeneous magnitude intensity patterns that vary across frequencies and attack types. Motivated by this observation, we introduce **MANI-Pure**, a magnitude-adaptive purification framework that leverages the magnitude spectrum of inputs to guide the purification process. Instead of injecting homogeneous noise, MANI-Pure adaptively applies heterogeneous, frequency-targeted noise, effectively suppressing adversarial perturbations in fragile high-frequency, low-magnitude bands while preserving semantically critical low-frequency content. Extensive experiments on CIFAR-10 and ImageNet-1K validate the effectiveness of MANI-Pure. It narrows the clean accuracy gap to within **0.59%** of the original classifier, while boosting robust accuracy by **2.15%**, and achieves the **top-1** robust accuracy on the RobustBench leaderboard, surpassing the previous state-of-the-art method.

## 1 INTRODUCTION

Deep neural networks have achieved remarkable success across diverse applications. However, their vulnerability to adversarial perturbations remains a critical challenge (Weng et al., 2023; Tao et al., 2024; Goodfellow et al., 2014), particularly in safety-critical domains where reliability is paramount (Bortsova et al., 2021; Shao et al., 2025; Ye et al., 2024). A primary line of defense is adversarial training (AT), which augments training with adversarial examples to enhance robustness (Mao et al., 2023; Schlarmann et al., 2024). Although effective, AT incurs substantial computational costs and suffers from limited generalization, posing challenges for both large-scale and cross-domain deployment. These limitations have motivated an alternative paradigm: adversarial purification (AP). Unlike AT, AP does not require retraining classifiers; instead purifies adversarial inputs at inference, restoring them to clean representations (Samangouei et al., 2018; Nie et al., 2022). This design offers flexibility, scalability, and compatibility with off-the-shelf models.

Diffusion-based purification (DBP) has become the most effective and widely adopted approach in AP. It suppresses perturbations by injecting uniform noise in the forward process and then reconstructing images via reverse diffusion. Several variants have been proposed, such as the gradual noise scheduling (Lee & Kim, 2023) and the purification-enhanced AT method (Lin et al., 2024).

Despite these advances, existing DBP and related defense methods often assume that adversarial perturbations are uniformly distributed across the frequency domain—an assumption that is contradicted by empirical evidence. As shown in Figure 1a, radial spectral analysis reveals that perturbations are unevenly concentrated in the high-frequency region. Figure 1b reflects the heterogeneity in magnitude intensity across different frequency bands and attack strategies. As a result, uniform noise injection faces a trade-off: strong noise disrupts low-frequency semantics, reducing clean accuracy, whereas weak noise fails to suppress high-frequency perturbations, thereby compromising robustness. This motivates the need for frequency-adaptive purification that targets perturbation-prone regions while preserving semantic fidelity.

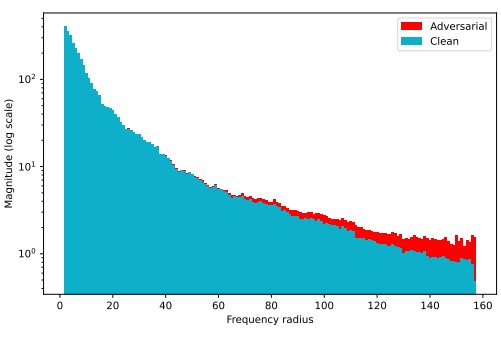 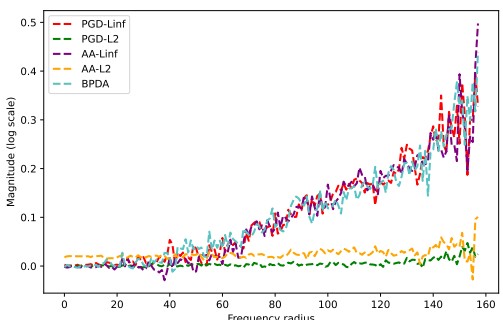

(a) Magnitude distribution differences between clean and adversarial images

(b) Noise magnitude spectra of common adversarial attacks

Figure 1: Radial spectrum analysis of adversarial perturbations. Overall, adversarial noise aligns with clean samples in low-to-mid frequencies but diverges in high-frequency bands. Specifically, **Left:** adversarial samples show irregular high-frequency peaks with uneven magnitude distribution. **Right:** common attacks concentrate perturbations in high-frequency regions, yet their spectral distributions and intensities differ significantly. These observations highlight the **limitation of uniform noise injection** and directly motivate our magnitude-adaptive design.

To address this challenge, we propose MANI-Pure, a magnitude-adaptive purification framework that redesigns the diffusion process from the frequency-domain perspective. The framework comprises two complementary modules:

- **MANI** adaptively adjusts the noise injection intensity across different regions based on the magnitude spectrum, ensuring the injected noise aligns with the vulnerability to perturbations while preserving the original image semantics from excessive distortion.

- **FreqPure** (Pei et al., 2025a) employs magnitude–phase decomposition to explicitly distinguish low and high frequency components, preserving low-frequency content while focusing purification on high frequencies.

Together, MANI emphasizes magnitude-aware adaptivity, while FreqPure enforces explicit frequency constraints. Their synergy enables precise suppression of concentrated perturbations while maximally retaining semantic structure, thereby improving robustness across diverse attacks.

We conduct extensive evaluations on CIFAR-10 (Krizhevsky et al., 2010) and ImageNet-1K (Deng et al., 2009) under strong adaptive attacks, including PGD+EOT (Madry et al., 2017; Athalye et al., 2018), AutoAttack (Croce & Hein, 2020), and BPDA+EOT (Hill et al., 2021). Results show that MANI-Pure significantly enhances robustness while maintaining high clean accuracy, consistently outperforming existing DBP methods. Importantly, the framework is plug-and-play, readily applicable to modern architectures such as CLIP (Radford et al., 2021), without additional training cost.

In summary, our main contributions are briefly summarized as follows:

- We empirically verify that adversarial perturbations are concentrated in high-frequency bands and further reveal **distributional differences** between adversarial and clean samples in the magnitude spectrum.

- The proposed MANI-Pure framework combines magnitude-adaptive diffusion with frequency-domain purification, achieving a principled balance between **semantic fidelity and perturbation mitigation**, reflected in improvements to both clean and robust accuracy.

- Extensive experiments across datasets, attacks, and backbones demonstrate the superiority of our method in terms of **robustness**, **clean accuracy** and **perceptual quality**, as well as its scalability as a **plug-and-play** module.

## 2 RELATED WORK

Adversarial purification provides a defense paradigm that restores adversarial inputs to clean representations at inference time, thereby avoiding the retraining cost of adversarial training.

**Generative Models for Adversarial Purification.** Early AP methods employed GANs, such as Defense-GAN (Samangouei et al., 2018), which projected adversarial samples onto the manifold of clean data. However, their limited generative fidelity and vulnerability to adaptive attacks significantly hindered their effectiveness. The advent of diffusion models marked a turning point: through stable likelihood-based training and high-quality reconstructions, they became the backbone of modern AP. Representative approaches include DiffPure (Nie et al., 2022), stochastic score-based denoising (Song et al., 2020), and gradient-guided purification like GDMP (Wang et al., 2022).

**Precision Noise Injection.** A key limitation of uniform noise injection lies in its disregard for the spectral structure of adversarial noise. Prior studies have shown that perturbations are often concentrate in high-frequency, low-magnitude regions (Yin et al., 2019). Building on this insight, FreqPure (Pei et al., 2025b) preserved low-frequency amplitude during reverse diffusion, effectively protects semantic content while targeting vulnerable high-frequency regions. These results highlight the importance of frequency-aware purification. Another line of research refines the forward noising process itself. Divide-and-Conquer (Pei et al., 2025a) integrates heterogeneous noise to better suppress adversarial perturbations, Sample-Specific Noise Injection (Sun et al., 2025) adapts noise to each input, and DiffCap (Fu et al., 2025) extends such ideas to vision–language models. While promising, these strategies remain largely fixed or heuristic, and they do not explicitly adapt to the actual spectral distribution of adversarial noise.

We unify these insights by introducing a magnitude-adaptive noise injection scheme that dynamically allocates noise to spectrally vulnerable regions, coupled with frequency-domain purification. This design enables precise suppression of perturbations while preserving semantic fidelity, thereby advancing AP toward finer-grained and more generalizable defenses.

## 3 METHODOLOGY

To eliminate adversarial perturbations while preserving semantic content, we propose **MANI-Pure**, a diffusion-based, frequency-domain purification framework comprising two complementary modules: **M**agnitude-**A**daptive **N**oise **I**njection (MANI) and **Freq**uency **Pur**ification (FreqPure). Figure 2 illustrates the overall structure. Before presenting the details, we briefly introduce the necessary background information.

### 3.1 PRELIMINARIES

We briefly introduce diffusion model, adversarial purification, and the frequency-domain theory relevant to our method.

**Diffusion Model.** Denoising Diffusion Probabilistic Model (DDPM) (Ho et al., 2020) generates data through a two-stage process: a forward noising process and a reverse denoising process.

*Forward process.* A sample $x_0$ is gradually perturbed into Gaussian noise through a Markov chain:

$$q(x_t \mid x_{t-1}) = \mathcal{N}\left(x_t; \sqrt{1 - \beta_t}\, x_{t-1}, \beta_t \mathbf{I}\right), \quad t = 1, \ldots, T, \tag{1}$$

where $\beta_t$ follows a predefined variance schedule. By marginalization:

$$q(x_t \mid x_0) = \mathcal{N}\left(x_t; \sqrt{\bar{\alpha}_t}\, x_0, (1 - \bar{\alpha}_t)\mathbf{I}\right), \tag{2}$$

with $\alpha_t = 1 - \beta_t$ and $\bar{\alpha}_t = \prod_{s=1}^{t} \alpha_s$.

*Reverse process.* To recover clean samples, the reverse distribution is approximated as

$$p_\theta(x_{t-1} \mid x_t) = \mathcal{N}\left(x_{t-1}; \mu_\theta(x_t, t), \sigma_t^2 \mathbf{I}\right). \tag{3}$$

Instead of predicting $\mu_\theta$ directly, DDPM parameterizes it with a noise predictor $\epsilon_\theta(x_t, t)$:

$$\mu_\theta(x_t, t) = \frac{1}{\sqrt{\bar{\alpha}_t}}\left(x_t - \frac{\beta_t}{\sqrt{1 - \bar{\alpha}_t}}\, \epsilon_\theta(x_t, t)\right), \tag{4}$$

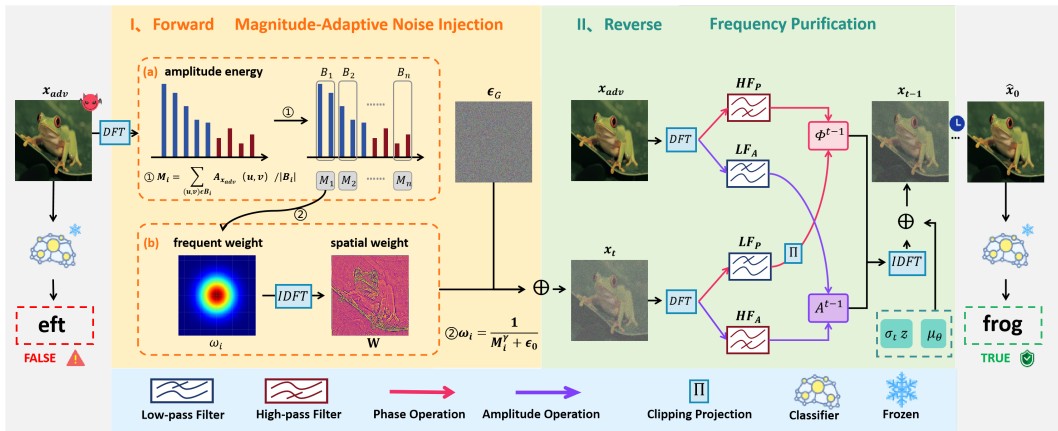

Figure 2: The pipeline of MANI-Pure. (I) **MANI**. Starting from an adversarial sample, we apply DFT to obtain its frequency representation, partition it into bands, compute average magnitudes, and derive band-wise and spatial weights. These weights modulate Gaussian noise to produce heterogeneous perturbations. (II) **FreqPure**. During the reverse process, the magnitude and phase spectra of the adversarial input and generated image are separated and recombined as shown, with the reconstructed image iteratively fed into subsequent denoising steps.

and the variance has a closed form:

$$\sigma_t^2 = \frac{1 - \bar{\alpha}_{t-1}}{1 - \bar{\alpha}_t} \beta_t. \tag{5}$$

*Sampling.* Starting from $x_T \sim \mathcal{N}(0, I)$, the model iteratively computes $x_{t-1} = \mu_\theta(x_t, t) + \sigma_t z$ with $z \sim \mathcal{N}(0, I)$ until $\hat{x}_0$ is obtained.

**Frequency-domain Theory.** For an image $x \in \mathbb{R}^{H \times W}$, the discrete Fourier transform (DFT) yields

$$\mathcal{F}(x)(u, v) = \sum_{h,w} x(h, w) e^{-2\pi i(uh/H + vw/W)}. \tag{6}$$

Each Fourier coefficient can be expressed in polar form as

$$\mathcal{F}(x)(u, v) = A_x(u, v) \cdot e^{i\Phi_x(u, v)}, \tag{7}$$

where $A_x(u, v) = |\mathcal{F}(x)(u, v)|$ is the magnitude spectrum, reflecting the intensity of frequency components, and $\Phi_x(u, v)$ is the phase spectrum, encoding structural and semantic information.

## 3.2 Magnitude-Adaptive Noise Injection

Building upon the frequency-domain preliminaries introduced in Section 3.1, we leverage the magnitude spectrum of the adversarial input $x_{\text{adv}}$ to capture the uneven distribution of frequency components. Specifically, the spectrum is partitioned into $n$ non-overlapping frequency bands $B_i$. The average magnitude in each band is computed as

$$M_i = \frac{1}{|B_i|} \sum_{(u,v) \in B_i} A_{x_{\text{adv}}}(u, v), \tag{8}$$

where $|B_i|$ denotes the number of coefficients in band $B_i$. This corresponds to step (a) of the magnitude-adaptive noise injection on the left in Figure 2.

Low-magnitude bands are empirically more vulnerable to adversarial perturbations, while high-magnitude bands correspond to dominant semantic structures. To emphasize fragile regions, we assign larger weights to lower-magnitude bands:

$$w_i = \frac{1}{M_i^\gamma + \epsilon_0}, \tag{9}$$

where $\gamma$ controls the sharpness of weighting and $\epsilon_0$ prevents numerical instability when $M_i$ is very small. The band-wise weights produce a frequency-domain weight distribution, which is transformed back to the spatial domain via inverse DFT to obtain a pixel-wise noise intensity map $\mathbf{W}$. In Figure 2, step (b) shows a visual representation of these two weights.

The spatial map $\mathbf{W}$ modulates Gaussian noise $\epsilon_G \sim \mathcal{N}(0, I)$ by element-wise multiplication:

$$\epsilon_t = \mathbf{W} \odot \epsilon_G, \quad \text{s.t. } \mathbf{W}, \epsilon_G \in \mathbb{R}^{H \times W \times C}. \tag{10}$$

Hence, the forward diffusion process becomes:

$$x_t = \sqrt{\bar{\alpha}_t}\, x_{\text{adv}} + \sqrt{1 - \bar{\alpha}_t}\, \epsilon_t, \tag{11}$$

where $\bar{\alpha}_t$ is the cumulative product of noise scheduling coefficients.

### 3.3 FREQUENCY PURIFICATION

To complement MANI, we further adopt a frequency purification strategy (Pei et al., 2025b) during the reverse diffusion process. The key observation is that low-frequency magnitude components exhibit strong robustness against adversarial perturbations, whereas the phase spectrum is more easily affected across all frequencies.

For an image $x_t$ generated during the reverse process, its DFT can be decomposed into magnitude $A_t$ and phase $\Phi_t$, with FreqPure handling them separately.

*Magnitude purification.* A low-pass filter $\mathcal{H}$ is applied to retain the low-frequency part of the adversarial input $x_{\text{adv}}$, while the high-frequency part is taken from the current generated image $x_t$:

$$A^{t-1} = \mathcal{H}(A_{\text{adv}}) + (1 - \mathcal{H})(A_t). \tag{12}$$

*Phase purification.* Low-frequency components are preserved through a projection operator $\Pi_\delta(\cdot)$ that restricts the generated phase within a small neighborhood of the adversarial phase:

$$\Phi^{t-1} = \mathcal{H}\big(\Pi_\delta(\Phi_t, \Phi_{\text{adv}})\big) + (1 - \mathcal{H})(\Phi_t), \tag{13}$$

where $\Pi_\delta(\Phi_t, \Phi_{\text{adv}})$ denotes clipping $\Phi_t$ into $[\Phi_{\text{adv}} - \delta,\ \Phi_{\text{adv}} + \delta]$, and $\delta$ is a hyperparameter controlling projection strength.

**Reconstruction.** The purified frequency representation $(A^{t-1}, \Phi^{t-1})$ is then transformed back into the spatial domain using the inverse discrete Fourier transform (IDFT):

$$x_{t-1} = \mathcal{F}^{-1}\left(A^{t-1}, \Phi^{t-1}\right), \tag{14}$$

and iteratively participates in the reverse diffusion process until $\hat{x}_0$ is obtained. The above process is described in the corresponding module on the right side of Figure 2.

Overall, FreqPure leverages the stability of low-frequency magnitudes while constraining the phase distribution, preventing structural distortions. In contrast, MANI avoids redundant noise in robust regions and focuses perturbations on vulnerable frequency bands, enabling effective denoising with minimal semantic loss. Together, they are complementary: MANI selectively **suppresses adversarial signals** in the forward process, while FreqPure ensures frequency stability and semantic consistency in the reverse process. The above methods are summarized in Appendix E.1.

## 4 EXPERIMENTS

### 4.1 EXPERIMENTAL SETUP

**Datasets and Model Architectures.** We conduct experiments on two widely used datasets of different resolutions: CIFAR-10 and ImageNet-1K. Following the settings in prior works (Pei et al., 2025a; Zhang et al., 2025b), we randomly select 512 samples from CIFAR-10 and 1,000 samples from ImageNet-1K for evaluation. To better align with the development of large-scale multimodal models, we adopt CLIP as the frozen classifier to accomplish zero-shot classification tasks. For the diffusion models, we use the publicly released unconditional CIFAR-10 checkpoint of EDM (Karras et al., 2022) for CIFAR-10, and 256x256 unconditional diffusion checkpoint for ImageNet-1K.

**Evaluation Metrics.** We report both standard accuracy and robust accuracy. This dual evaluation provides a comprehensive view of the trade-off between preserving performance on clean data and enhancing resilience against attacks.

**Attack Settings.** In our experiments, we evaluate all defenses under strong adaptive attacks across both $\ell_\infty$ and $\ell_2$ threat models. Concretely, we employ PGD and AutoAttack as primary evaluation tools, covering both $\ell_\infty$ and $\ell_2$ perturbations. Following Lee & Kim (2023), we adopt PGD combined with expectation over transformations (PGD+EOT) to mitigate variability caused by stochastic components in the defense. In addition, we test BPDA+EOT to evaluate attacks that approximate gradients through non-differentiable or randomized components. For computational tractability while retaining attack strength, PGD and BPDA are run for 10 iterations, and EOT uses 10 samples per gradient estimate. AutoAttack is executed in its standard version. The perturbation budgets are specified as $\epsilon = 8/255$ for $\ell_\infty$ attacks on CIFAR-10, $\epsilon = 4/255$ for $\ell_\infty$ attacks on ImageNet, and $\epsilon = 0.5$ for $\ell_2$ attacks on both datasets. Further experimental settings can be found in Appendix D.

## 4.2 MAIN RESULTS

This section presents a comprehensive evaluation of MANI-Pure across multiple datasets, attack settings, and metrics, with a focus on **robustness**, **perceptual quality**, and **plug-and-play flexibility**.

### 4.2.1 CLASSIFICATION ACCURACY UNDER ADAPTIVE ATTACKS

**MANI-Pure consistently achieves the best trade-off between standard and robust accuracy across datasets and backbones.** As summarized in Table 1 (CIFAR-10, ViT-L/14), Table 2 (CIFAR-10, RN50), and Table 3 (ImageNet-1K, ViT-L/14), we evaluate against strong adaptive attacks including PGD+EOT, AutoAttack under both $\ell_\infty$ and $\ell_2$ norms, and BPDA+EOT.

On CIFAR-10, MANI-Pure improves robust accuracy by **2.15%** under AutoAttack ($\ell_\infty$) and by **2.54%** under BPDA+EOT when using ViT-L/14. Consistent improvements are also observed on RN50, confirming the backbone-agnostic nature of our framework. On ImageNet-1K, especially, MANI-Pure achieves the highest robust accuracy, outperforming all baselines by **3.8%** under BPDA+EOT, while maintaining competitive clean accuracy.

These results demonstrate that MANI-Pure not only surpasses existing AP and AT baselines (including recent leaders on RobustBench), but also exhibits strong cross-dataset generalization and backbone versatility. More results on different backbones can be found in the Appendix. E.2.

Table 1: Classification accuracy on CIFAR-10 under adversarial attacks using CLIP ViT-L/14. Zero-shot CLIP (w/o defense) is denoted by †, its standard accuracy as the upper bound. Methods from the Robustbench leaderboard are denoted by ‡. AT and AP methods are marked accordingly.

| Type | Algorithm | Standard | PGD | | AutoAttack | | BPDA |
|---|---|---|---|---|---|---|---|
| | | | $\ell_\infty$ | $\ell_2$ | $\ell_\infty$ | $\ell_2$ | |
| **AT** | *DHAT* (Zhang et al., 2025a) | 85.45 | 63.14 | 66.91 | 56.77 | 57.40 | 54.84 |
| | *DIAT* (Wang et al., 2023) ‡ | 92.69 | 71.38 | 85.12 | 70.53 | 84.03 | 69.76 |
| | *MeanSparse* (Amini et al., 2024) ‡ | 92.98 | 74.02 | 86.41 | 68.85 | 85.98 | 72.87 |
| **AP** | Zero-shot (w/o defense)† | 94.73 | 2.15 | 55.86 | 0.00 | 0.00 | 0.78 |
| | + *DiffPure* (Nie et al., 2022) | 86.52 | 85.55 | 85.74 | 85.35 | 85.55 | 84.96 |
| | + *DDPM++* (Song et al., 2020) | 86.33 | 84.77 | 85.16 | 85.74 | 85.74 | 86.13 |
| | + *REAP* (Lee & Kim, 2023) | 81.45 | 79.69 | 79.87 | 80.08 | 80.18 | 80.86 |
| | + *FreqPure* (Pei et al., 2025b) | 91.77 | 90.17 | 91.41 | 90.82 | 91.99 | 87.89 |
| | + *CLIPure* (Zhang et al., 2025b) | 93.55 | 89.06 | 92.19 | 90.04 | 92.38 | 83.01 |
| | + *Ours* | **94.14** | **91.02** | **92.58** | **92.19** | **93.16** | **88.67** |

Table 2: Classification accuracy on CIFAR-10 under adversarial attacks using CLIP RN50. Zero-shot CLIP (w/o defense) is denoted by †, its standard accuracy as the upper bound. Only AP-based methods are included.

| Algorithm | Standard | PGD | | AutoAttack | | BPDA |
|---|---|---|---|---|---|---|
| | | $\ell_\infty$ | $\ell_2$ | $\ell_\infty$ | $\ell_2$ | |
| Zero-shot (w/o defense)† | 69.92 | 0.00 | 19.73 | 0.39 | 0.39 | 3.32 |
| + *DiffPure* (Nie et al., 2022) | 61.91 | 59.77 | 61.13 | 59.77 | 60.64 | 60.16 |
| + *DDPM++* (Song et al., 2020) | 56.64 | 56.25 | 56.64 | 56.05 | 56.34 | 55.27 |
| + *REAP* (Lee & Kim, 2023) | 58.59 | 56.84 | 58.40 | 55.66 | 58.40 | 56.25 |
| + *FreqPure* (Pei et al., 2025b) | 62.70 | 59.38 | 60.55 | 61.52 | 62.56 | 58.79 |
| + *CLIPure* (Zhang et al., 2025b) | 61.33 | 53.71 | 60.55 | 56.84 | 60.55 | 53.32 |
| *+Ours* | **65.23** | **61.91** | **62.50** | **62.70** | **64.84** | **60.16** |

Table 3: Classification accuracy on ImageNet-1K under adversarial attacks using CLIP ViT-L/14. Zero-shot CLIP (w/o defense) is denoted by †, its standard accuracy as the upper bound. Only AP-based methods are included.

| Algorithm | Standard | PGD | | AutoAttack | | BPDA |
|---|---|---|---|---|---|---|
| | | $\ell_\infty$ | $\ell_2$ | $\ell_\infty$ | $\ell_2$ | |
| Zero-shot (w/o defense)† | 74.90 | 1.20 | 31.60 | 0.10 | 0.10 | 0.00 |
| + *DiffPure* (Nie et al., 2022) | 71.10 | 43.00 | 43.40 | 42.90 | 44.20 | 42.50 |
| + *DDPM++* (Song et al., 2020) | 70.70 | 66.00 | 70.00 | 68.10 | 70.40 | 63.50 |
| + *REAP* (Lee & Kim, 2023) | 51.30 | 48.90 | 49.90 | 48.40 | 50.10 | 48.50 |
| + *OSCP* (Lei et al., 2025) | 71.60 | 65.70 | 69.00 | 68.30 | 70.10 | 66.00 |
| + *Ours* | **73.10** | **67.30** | **70.80** | **68.90** | **70.90** | **67.30** |

### 4.2.2 PERCEPTUAL QUALITY EVALUATION

**MANI-Pure produces purified images that are perceptually closest to clean images across different backbones.** Since diffusion-based purification is inherently generative, we complement robustness evaluation with perceptual quality metrics, conducted on the CIFAR-10 dataset. Table 4 reports results on SSIM (Wang et al., 2004) (higher is better) and LPIPS (Zhang et al., 2018) (lower is better). On RN50, MANI-Pure achieves an SSIM of **0.9274** and an LPIPS of **0.1136**, both outperforming all baselines. Similar trends are observed with ViT-L/14. Overall, MANI-Pure consistently achieves the highest perceptual similarity, underscoring its ability to defend against adversarial perturbations while preserving image fidelity.

Table 4: To evaluate the quality of the generated images, we compute the SSIM and LPIPS scores between the images purified by different AP methods and the clean images.

| Backbone | Metric | Methods | | | | |
|---|---|---|---|---|---|---|
| | | Adversarial | DiffPure | REAP | FreqPure | Ours |
| ViT-L/14 | SSIM ↑ | 0.8204 | 0.8342 | 0.8044 | 0.9172 | **0.9270** |
| | LPIPS↓ | 0.4403 | 0.2110 | 0.2553 | 0.1214 | **0.1133** |
| RN50 | SSIM ↑ | 0.8180 | 0.8344 | 0.8045 | 0.9176 | **0.9274** |
| | LPIPS↓ | 0.3907 | 0.2110 | 0.2551 | 0.1217 | **0.1136** |

### 4.2.3 QUALITATIVE VISUALIZATION

**Visualizations confirm that MANI-Pure selectively suppresses adversarial perturbations while preserving semantics.** To better validate the effectiveness of adaptive noise injection, we visual-

ize the difference between the injected noise and adversarial noise. Figure 3 clearly shows that adaptive noise aligns much better with adversarial perturbations than uniform noise, especially in high-frequency regions that are most vulnerable to attacks. Quantitatively, KL divergence further confirms this observation: adaptive noise (**0.1628**) is substantially closer to adversarial noise than uniform noise (**0.4988**).

These findings highlight our core advantage—precise suppression of adversarially vulnerable regions while preserving semantic fidelity elsewhere.

We further compare purified samples from DDPM++ and MANI-Pure, together with pixel-wise difference heatmaps relative to clean images. Our method introduces smaller modifications in low-frequency background regions, while applying targeted changes in high-frequency regions most affected by perturbations, which provides direct evidence of MANI-Pure's frequency-adaptive design.

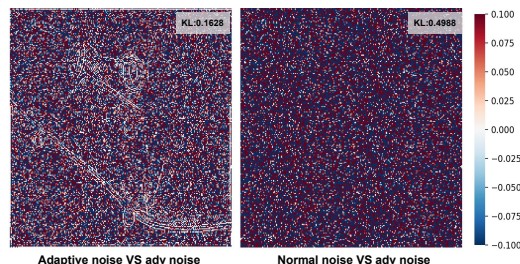

Figure 3: Difference heatmaps between adaptive noise (left) / uniform noise (right) and adversarial noise. Lighter colors indicate smaller differences.

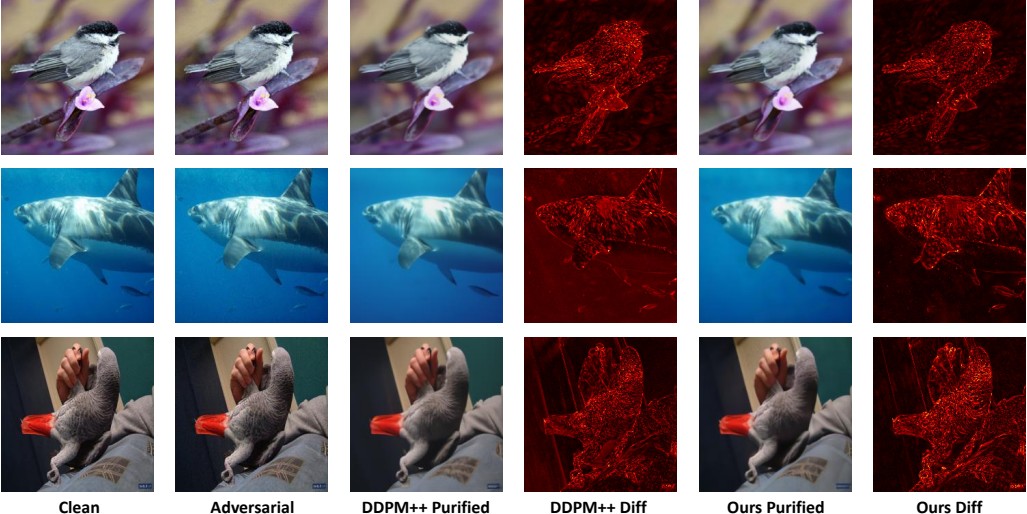

Figure 4: Visualization of purification before and after defense. The figure compares purified results from DDPM++ and MANI-Pure, together with pixel-wise difference heatmaps relative to clean images. **Overall:** MANI-Pure introduces smaller modifications in low-frequency background regions, avoiding unnecessary semantic loss. **Key effect:** it selectively alters high-frequency vulnerable regions, providing direct evidence of its frequency-adaptive design.

### 4.2.4 PLUG-AND-PLAY COMPATIBILITY

As a modular noise injection strategy, MANI can be seamlessly combined with various existing DBP methods. Table 5 reports results under $\ell_\infty$ attacks (results under $\ell_2$ are listed in Appendix E.3).

We observe that **MANI consistently improves both clean and robust accuracy** across all tested AP baselines. In particular, REAP benefits the most, with its clean accuracy increased by **4.10%** and robust accuracy under AutoAttack improved by **1.95%**. More importantly, the combination of MANI with FreqPure yields the overall best performance, highlighting the **complementary design philosophy** between the two modules. These results validate MANI as a general and effective plug-in for enhancing diverse purification pipelines.

Table 5: **Plug-and-play validation of the MANI module under $\ell_\infty$ attacks.** We integrated MANI into various diffusion-based purification frameworks and evaluated them on CIFAR-10. Results are reported both without MANI (**w/o**) and with MANI (**w/**).

| Algorithm | Standard | | PGD | | AutoAttack | | BPDA | |
|---|---|---|---|---|---|---|---|---|
| | w/o | w/ | w/o | w/ | w/o | w/ | w/o | w/ |
| +*DiffPure* (Nie et al., 2022) | 86.52 | 87.72 | 85.55 | 86.82 | 85.35 | 86.91 | 84.96 | 85.43 |
| +*DDPM++* (Song et al., 2020) | 86.33 | 87.30 | 84.77 | 86.33 | 85.74 | 86.52 | 86.13 | 86.33 |
| +*REAP* (Lee & Kim, 2023) | 81.45 | 85.55 | 79.69 | 81.45 | 80.08 | 82.03 | 80.86 | 82.42 |
| +*FreqPure* (Pei et al., 2025b) | 91.77 | 94.14 | 90.17 | 91.02 | 90.82 | 92.19 | 87.89 | 88.67 |

## 4.3 ABLATION STUDIES

We conducted ablation experiments on CIFART-10 to better understand the contributions of different design choices in MANI-Pure, primarily involving parameter analysis and module ablation.

**Effect of hyperparameters.** The MANI module mainly involves two hyperparameters: the weighting factor $\gamma$ and the number of frequency bands $n$. As shown in Figure 5, both standard and robust accuracy exhibit a "rise-then-fall" trend as $\gamma$ increases from 1.0. Specifically, standard accuracy peaks at $\gamma = 1.6$, while $\gamma = 1.8$ achieves a more balanced trade-off between clean and robust performance. A similar trend is observed for $n$, where $n = 8$ provides the best overall results.

**Effect of different modules.** To further assess the contribution of each component, we conduct ablation studies on MANI and FreqPure. As shown in Table 6, both modules individually enhance the baseline performance. When combined, they yield substantially larger improvements than using either module alone, achieving gains of 7.62% in clean accuracy and 5.47% in robust accuracy. These results highlight the orthogonal benefits of MANI and FreqPure, and their strong complementarity.

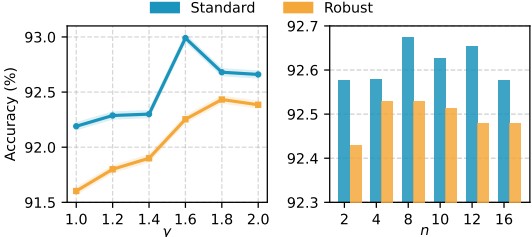

Figure 5: Standard accuracy and robust accuracy under different ratio factor $\gamma$ (left) and under different number of frequency band $n$ (right).

Table 6: Standard and robust accuracy for different block combinations. ✓ and ✗ indicate use or non-use of the module.

| MANI | FreqPure | Standard | Robust |
|---|---|---|---|
| ✗ | ✗ | 86.52 | 85.55 |
| ✓ | ✗ | 87.30 | 86.33 |
| ✗ | ✓ | 91.77 | 90.17 |
| ✓ | ✓ | **94.14** | **91.02** |

## 5 CONCLUSION

This work systematically analyzes the distribution of adversarial perturbations in the frequency domain and shows that existing uniform noise injection strategies may disrupt the semantic structure of clean images. To address this issue, we propose **MANI-Pure**, a diffusion-based purification framework that integrates magnitude-adaptive noise injection to emphasize vulnerable frequency bands and frequency purification to protect semantic structures. Through extensive experiments on two benchmark datasets under multiple attacks, MANI-Pure effectively suppresses adversarial noise while preserving semantic content, achieving a favorable balance between clean and robust accuracy. Moreover, the plug-and-play design of MANI highlights its compatibility with diverse purification pipelines, further broadening its applicability.

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

APPENDIX OVERVIEW

This appendix provides additional details and analyses to complement the main paper. It is organized as follows:

- **Section A. Use of Large Language Models.** We clarify the extent to how LLMs were used during the writing and proofreading process, ensuring transparency in compliance with conference policies.
- **Section B. Background on Adversarial Attacks and Defenses.** We review standard adversarial attacks (e.g., PGD, AutoAttack, BPDA) and defense paradigms (adversarial training, purification), offering context for how our method relates to existing approaches.
- **Section C. Theoretical Supplement.** We provide a more complete derivation of diffusion models, present a unified mathematical framework for adversarial purification, and analyze the computational complexity and stability of different approaches.
- **Section D. Experimental Settings.** We detail the hyperparameter choices for both attacks and diffusion models, including perturbation budgets, iteration numbers, noise schedules, and pretrained checkpoints, ensuring reproducibility of all results.
- **Section E. Additional Experimental Results.** We extend the evaluations beyond the main text. This includes: (i) a step-by-step algorithmic workflow of our framework. (ii) classification with alternative backbones (CLIP-RN101, WRN-28-10,RN-50), (iii) plug-and-play integration under $\ell_2$ attacks, (iv) analysis of PGD iteration numbers, and
- **Section F. Visualization.** We provide additional qualitative results, showing purified versus adversarial samples across multiple datasets, highlighting the semantic preservation and noise suppression of our method.

## A   STATEMENT ON THE USE OF LLMS

This study employed LLMs to assist in writing. LLMs were primarily utilized for language refinement, grammatical corrections, and enhancing academic tone. It is crucial to emphasize that all viewpoints, theoretical frameworks, experimental results, and final conclusions were independently developed by human authors. LLMs served solely as auxiliary tools for manuscript refinement, with all final drafts thoroughly reviewed and approved by the authors.

## B   SUPPLEMENT RELATED WORK

**Adversarial Attacks & Robustness.** Adversarial attacks have long revealed the fragility of neural networks, beginning with the discovery of imperceptible perturbations by Szegedy et al. (2013) and the efficient one-step FGSM attack (Goodfellow et al., 2014). Iterative methods such as PGD (Madry et al., 2017) established strong benchmarks for robustness evaluation, later extended by efficient variants like FreeAT (Shafahi et al., 2019) and AutoAttack (Croce & Hein, 2020). The use of EOT (Expectation over Transformation) (Athalye et al., 2018) was further emphasized to mitigate randomness and non-differentiability in gradients, ensuring accurate robustness assessment. On the defense side, adversarial training (Schlarmann et al., 2024; Mao et al., 2023) remains the most widely used strategy. By incorporating adversarial examples into the training process, AT explicitly improves the decision boundary against perturbations, thereby enhancing robustness. However, AT requires significant computational resources and often generalizes poorly to unseen attacks, motivating research into alternative approaches.AP emerged in response to this situation.

## C   THEORETICAL SUPPLEMENT

### C.1   UNIFIED FRAMEWORK FOR ADVERSARIAL PURIFICATION

We can unify diffusion-based adversarial purification methods into the following generalized formulation:

$$x_t = f(x_0; \bar{\alpha}_t) + g(\epsilon; \mathbf{W}), \tag{15}$$

where $f(x_0; \bar{\alpha}_t) = \sqrt{\bar{\alpha}_t}\, x_0$ denotes the signal decay term, $g(\epsilon; \mathbf{W})$ represents noise injection, and $\mathbf{W}$ is a weighting or transformation operator.

- **Adversarial Training:** robustness stems from model parameters; no explicit $g(\cdot)$ is introduced.
- **DiffPure:** $g(\epsilon; \mathbf{W}) = \sqrt{1 - \bar{\alpha}_t}\, \epsilon$, where $\mathbf{W} = I$.
- **MANI-Pure:** $g(\epsilon; \mathbf{W}) = \sqrt{1 - \bar{\alpha}_t}(\mathbf{W} \odot \epsilon)$, where $\mathbf{W}$ is derived from frequency magnitudes.
- **FreqPure:** constraints are imposed in the *reverse* step, by spectral recombination rather than forward-side weighting.

This unified framework highlights a key dichotomy: *forward-side approaches* redesign $g(\cdot)$ to better mimic adversarial distributions, while *reverse-side approaches* constrain the reconstruction trajectory. MANI-Pure naturally combines both perspectives, explaining its superior performance.

## C.2 COMPLEXITY AND STABILITY ANALYSIS

**Time Complexity:**

- **DiffPure:** $O(T \cdot HW)$ per reverse trajectory, dominated by neural network inference.
- **MANI-Pure:** adds DFT/IDFT operations of $O(HW \log(HW))$ per step, negligible compared to network cost.
- **FreqPure:** incurs extra spectral recombination and projection, but all operations are element-wise or FFT-based, remaining parallelizable on GPUs.
- **Hybrid methods (e.g., MANI+FreqPure):** maintain linear scaling in $T$ and near-constant overhead relative to the diffusion backbone.

**Space Complexity:**

- All methods store $O(HW)$ activations per step.
- Frequency-based approaches require one additional complex-valued copy of the spectrum, i.e., $O(2HW)$, which is marginal compared with feature maps inside the denoiser.

**Numerical Stability:**

- FFT and inverse FFT are unitary transforms, introducing no instability.
- MANI's band-wise weighting may amplify small magnitudes, but normalization with $\epsilon$ ensures bounded variance.
- FreqPure's projection operator $\Pi(\cdot)$ restricts phase drift, effectively stabilizing the reverse trajectory under strong attacks.

**Scalability.** Since the extra overhead scales sub-linearly with resolution ($\log(HW)$), frequency-domain operations remain efficient even for high-resolution ImageNet-1K images. Therefore, the proposed MANI-Pure achieves robustness gains without sacrificing efficiency.

## D PARAMETERS AND SETTINGS

### D.1 ATTACK SETUP

We adopt three types of strong adaptive attacks: PGD+EOT, AutoAttack, and BPDA+EOT. For PGD and BPDA, the number of iterations is set to 10 (the rationale for this choice is discussed in Appendix E.4), while the number of EOT samples is also set to 10. AutoAttack is executed in its standard version, which integrates `APGD-CE`, `APGD-DLR`, `FAB`, and `Square Attack`, with 100 update iterations. The perturbation budget is $\epsilon = 8/255$ for $\ell_\infty$ attacks on CIFAR-10 and $\epsilon = 4/255$ on ImageNet-1K, while $\ell_2$ attacks use $\epsilon = 0.5$ for both datasets. Unless otherwise specified, the step size is set to 0.007 for all attacks.

## D.2 DIFFUSION SETUP

Our purification framework is based on DDPM++ (Song et al., 2020) with a linear variance schedule, where the noise variance increases from $\beta_1 = 10^{-4}$ to $\beta_T = 0.02$ over $T = 1000$ steps (Ho et al., 2020). In all experiments, we set the forward noising steps to 100 and the reverse denoising steps to 5, unless otherwise specified. For DiffPure, we follow the original implementation and use 100 reverse steps. The pretrained diffusion weights are taken from public releases: the unconditional CIFAR-10 checkpoint of EDM (Karras et al., 2022) and the $256 \times 256$ unconditional diffusion checkpoint for ImageNet-1K, consistent with prior works.

## D.3 NOISE DIFFERENCE HEATMAP COMPUTATION

To analyze the similarity between injected noise $N_{\mathrm{inj}}$ and adversarial noise $N_{\mathrm{adv}}$, we compute their pixel-wise difference:

$$D = N_{\mathrm{inj}} - N_{\mathrm{adv}}. \tag{16}$$

Here $D$ contains both positive and negative values, where the sign indicates whether the injected noise is larger or smaller than the adversarial noise at each pixel. For visualization, we normalize $D$ and render it with a diverging colormap, where red/blue colors represent positive/negative differences, respectively.

# E ADDITIONAL RESULTS

## E.1 THE ALGORITHM WORKFLOW OF MANI-PURE

This section presents the **MANI-Pure** algorithm flowchart (Algorithm 1), which comprehensively illustrates the entire processing workflow. This contrasts with the section-by-section module introductions in Sec. 3.2 and the abstract representation in Figure 2.

---

**Algorithm 1** Adversarial Purification with MANI and FreqPure

---

**Require:** Adversarial input $x_{\mathrm{adv}}$, Diffusion steps $T$, Band number $n$, Weighting factor $\gamma$
**Ensure:** Purified image $x_0$
1: $(A_{\mathrm{adv}}, \Phi_{\mathrm{adv}}) = \mathcal{F}(x_{\mathrm{adv}})$
2: Partition $M_{\mathrm{adv}}$ into $n$ frequency bands $\{B_i\}$          // Forward Progress:MANI
3: **for** each band $B_i$ **do**
4:     $M_i = \frac{1}{|B_i|} \sum_{(u,v) \in B_i} A_{\mathrm{adv}}(u,v)$
5:     $w_i = (M_i + \epsilon_0)^{-\gamma}$
6: **end for**
7: Construct spatial weight map $W$ via IDFT
8: $\epsilon_t = W \odot \epsilon_G$, with $\epsilon_G \sim \mathcal{N}(0, I)$
9: $x_t = \sqrt{\bar{\alpha}_t}\, x_{\mathrm{adv}} + \sqrt{1 - \bar{\alpha}_t}\, \epsilon_t$
10: Initialize $x_T \sim \mathcal{N}(0, I)$          // Reverse Progress:FreqPure
11: **for** $t = T \to 1$ **do**
12:     $x_{0|t} = \frac{1}{\sqrt{\alpha_t}}\big(x_t - \sqrt{1 - \bar{\alpha}_t}\, \epsilon_\theta(x_t, t)\big)$
13:     $(A_t, \Phi_t) = \mathcal{F}(x_{0|t})$
14:     $A^{t-1} = \mathcal{H}(A_{\mathrm{adv}}) + (1 - \mathcal{H})(A_t)$
15:     $\Phi^{t-1} = \mathcal{H}\big(\Pi(\Phi_t, \Phi_{\mathrm{adv}}, \delta)\big) + (1 - \mathcal{H})(\Phi_t)$
16:     $x_{t-1} = \mathcal{F}^{-1}(A^{t-1}, \Phi^{t-1})$
17: **end for**
18: **return** $x_0$

---

## E.2 ROBUSTNESS UNDER DIFFERENT BACKBONES

In this section, we further supplement classification experiments with CLIP (RN101), WRN-28-10 (Zagoruyko & Komodakis, 2016) and ResNet-50 (He et al., 2016), following the same settings as Sec. 4.1 in the main text. As shown in Table 1, Table 2, Table 7, Table 8 and Table 9, **MANI-Pure**

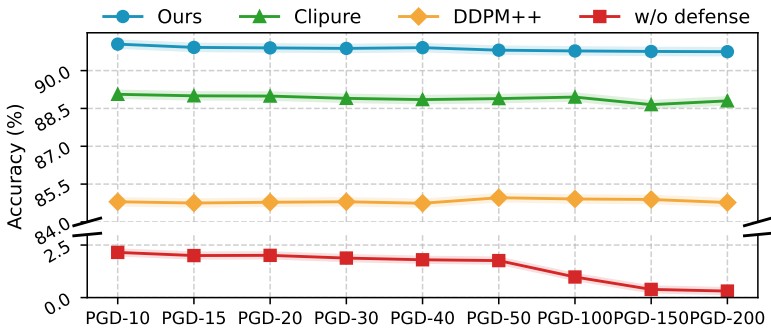

Figure 6: Robust accuracy of several purification methods across different PGD iteration counts (All attacks with EOT=10).

**consistently achieves the best performance across different classifier architectures**, demonstrating its versatility and robustness.

Table 7: Classification accuracy on CIFAR-10 under adversarial attacks using CLIP RN101. Zero-shot CLIP (w/o defense) is denoted by †; its standard accuracy as the upper bound. Only AP-based methods are included.

| Algorithm | Standard | PGD | | AutoAttack | | BPDA |
| --- | --- | --- | --- | --- | --- | --- |
| | | $\ell_\infty$ | $\ell_2$ | $\ell_\infty$ | $\ell_2$ | |
| Zero-shot (w/o defense) † | 78.32 | 0.00 | 26.56 | 0.20 | 0.20 | 2.73 |
| + *DiffPure* (Nie et al., 2022) | 67.58 | 65.98 | 66.60 | 65.62 | 66.60 | 66.01 |
| + *DDPM++* (Song et al., 2020) | 68.95 | 65.62 | 66.99 | 64.45 | 66.80 | 65.62 |
| + *REAP* (Lee & Kim, 2023) | 62.30 | 61.33 | 61.72 | 61.91 | 61.13 | 61.91 |
| + *FreqPure* (Pei et al., 2025b) | 70.70 | 68.55 | 68.95 | 67.97 | 68.75 | 66.80 |
| + *CLIPure* (Zhang et al., 2025b) | 68.95 | 62.89 | 68.75 | 64.26 | 68.84 | 59.18 |
| *+Ours* | **71.88** | **68.75** | **70.12** | **69.43** | **70.12** | **69.53** |

### E.3 PLUG-AND-PLAY RESULTS UNDER $\ell_2$ ATTACKS

In addition to the $\ell_\infty$ setting reported in the main text, we also evaluate the plug-and-play integration of MANI with existing AP methods under $\ell_2$ attacks. Following the same configurations as Sec. 4.1, we consider PGD+EOT and AutoAttack with perturbation budget $\epsilon = 0.5$. The results, summarized in Table 10, show that MANI consistently improves both clean and robust accuracy when combined with different AP backbones.

### E.4 EFFECT OF ATTACK ITERATIONS

We also examine the impact of the number of PGD iterations on robust accuracy. In our main experiments, we set PGD iterations to 10. Since prior works adopt different iteration counts, we perform an ablation to validate this choice. As illustrated in Figure 6, the robust accuracy of undefended models decreases sharply with more iterations and converges near zero, while defense methods remain relatively stable with only minor fluctuations. Therefore, we adopt 10 iterations as a practical **balance between robustness evaluation and computational efficiency**. Additionally, for EOT iterations, we follow the setting in Nie et al. (2022), which shows that robustness converges once EOT exceeds 10.

Table 8: Classification accuracy on CIFAR-10 under adversarial attacks using WRN-28-10. WRN-28-10(w/o defense) is denoted by †; its standard accuracy as the upper bound. Results marked with ‡ are reported in Bai et al. (2024). Only AP-based methods are included.

| Algorithm | Standard | PGD | AutoAttack |
|---|---|---|---|
| WRN-28-10 (w/o defense) † | 96.48 | 0.00 | 0.00 |
| +*Diffpure*(Nie et al., 2022) | 90.07 | 56.84 | 63.30 |
| +*REAP*(Lee & Kim, 2023) | 90.16 | 55.82 | 70.47 |
| +*CGDM*(Bai et al., 2024)‡ | 91.41 | 49.22 | 77.08 |
| +*FreqPure*(Pei et al., 2025b) | 92.19 | 59.39 | 77.35 |
| +*Ours* | **92.57** | **61.32** | **78.69** |

Table 9: Classification accuracy on CIFAR-10 under adversarial attacks using ResNet-50. ResNet-50(w/o defense) is denoted by †; its standard accuracy as the upper bound. Results marked with ‡ are reported in Bai et al. (2024). Only AP-based methods are included.

| Algorithm | Standard | PGD | AutoAttack |
|---|---|---|---|
| ResNet-50 (w/o defense) † | 76.01 | 0.00 | 0.00 |
| +*Diffpure*(Nie et al., 2022) | 67.84 | 42.58 | 41.53 |
| +*REAP*(Lee & Kim, 2023) | 68.72 | 43.19 | 44.67 |
| +*CGDM*(Bai et al., 2024)‡ | 68.98 | 41.80 | - |
| +*FreqPure*(Pei et al., 2025b) | 69.53 | 59.77 | **63.49** |
| +*Ours* | **70.31** | **60.03** | 61.79 |

Table 10: **Plug-and-play validation of the MANI module under $\ell_2$ attacks.** We integrated MANI into various diffusion-based purification frameworks and evaluated them on CIFAR-10. Results are reported both without MANI (**w/o**) and with MANI (**w/**).

| Algorithm | PGD | | AutoAttack | |
|---|---|---|---|---|
| | w/o | w/ | w/o | w/ |
| + *DiffPure* (Nie et al., 2022) | 85.74 | 87.08 | 85.55 | 87.50 |
| + *DDPM++* (Song et al., 2020) | 85.16 | 86.72 | 85.74 | 87.11 |
| + *REAP* (Lee & Kim, 2023) | 79.87 | 81.64 | 80.18 | 81.84 |
| + *FreqPure* (Pei et al., 2025b) | 91.41 | 92.58 | 92.00 | 93.16 |

## F  VISUALIZATION

To intuitively illustrate the purification effect, we present qualitative results on randomly selected samples from CIFAR-10 (Figure 7, Figure 8, Figure 9) and ImageNet-1K (Figure 10, Figure 11, Figure 12), including clean images, adversarial images, and purified images.

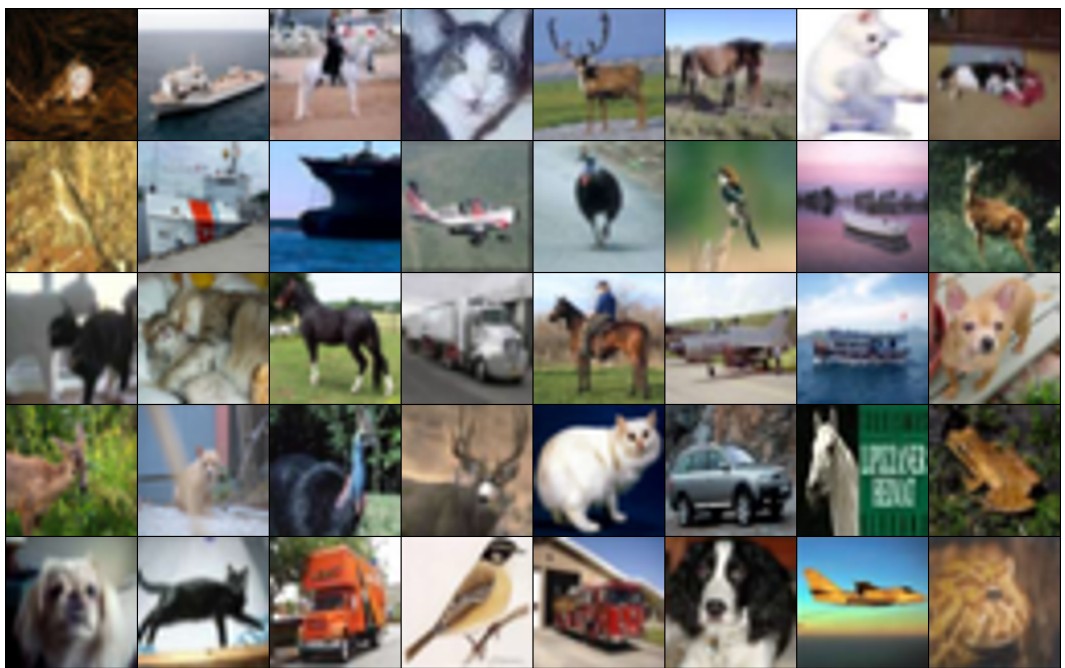

Figure 7: **Clean** CIFAR-10 images randomly selected for visualization

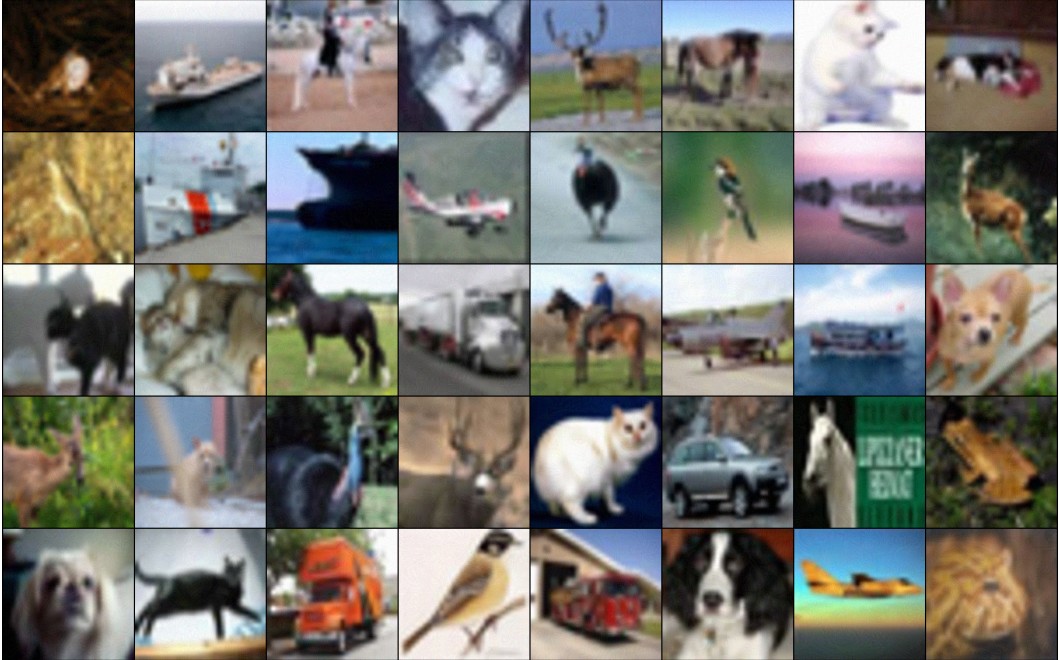

Figure 8: **Adversarial** CIFAR-10 images randomly selected for visualization

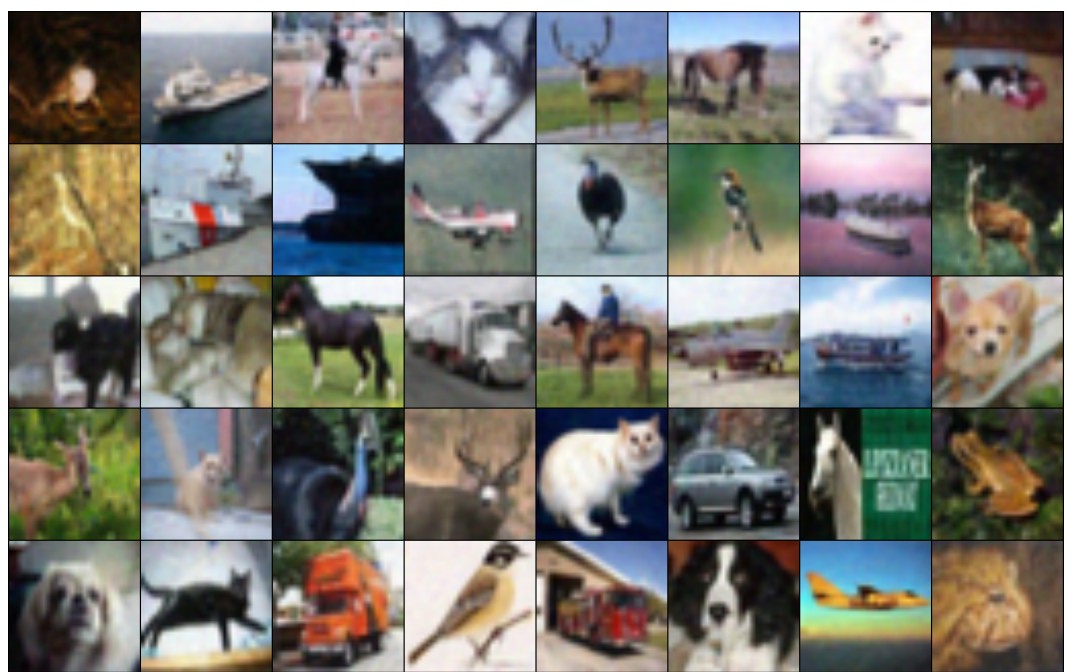

Figure 9: **Purified** CIFAR-10 images randomly selected for visualization

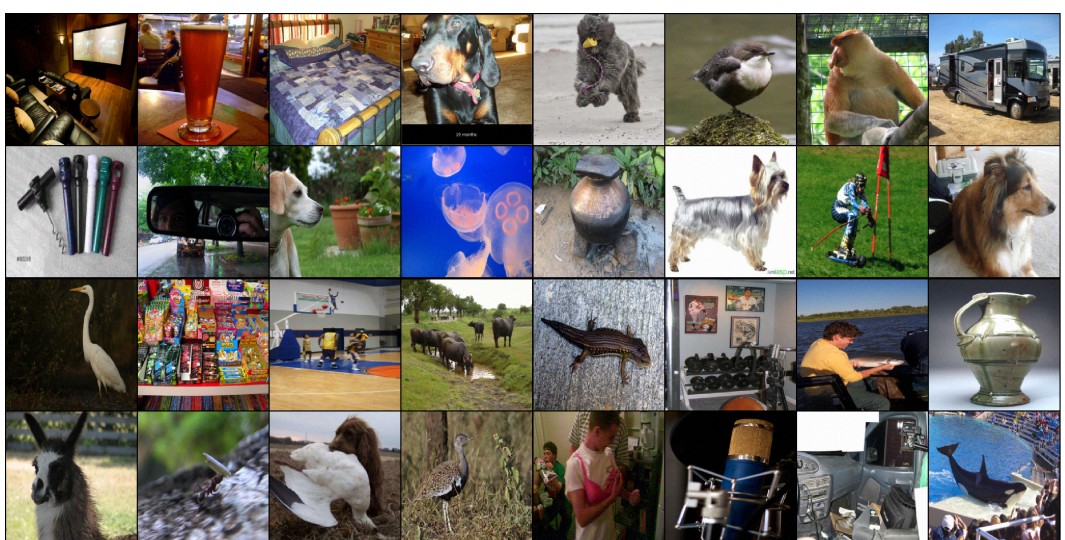

Figure 10: **Clean** ImageNet-1K images randomly selected for visualization

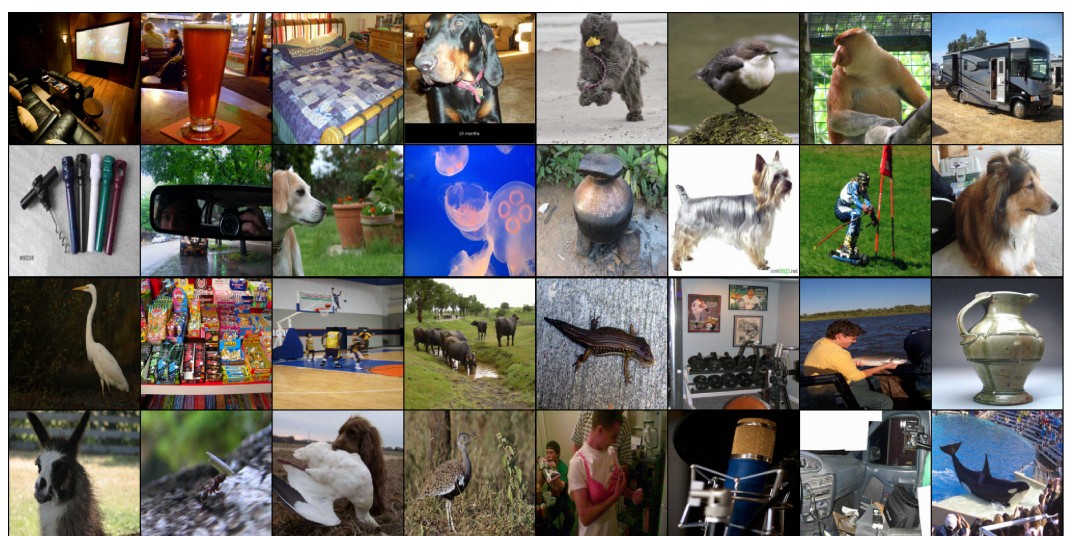

Figure 11: **Adversarial** ImageNet-1K images randomly selected for visualization

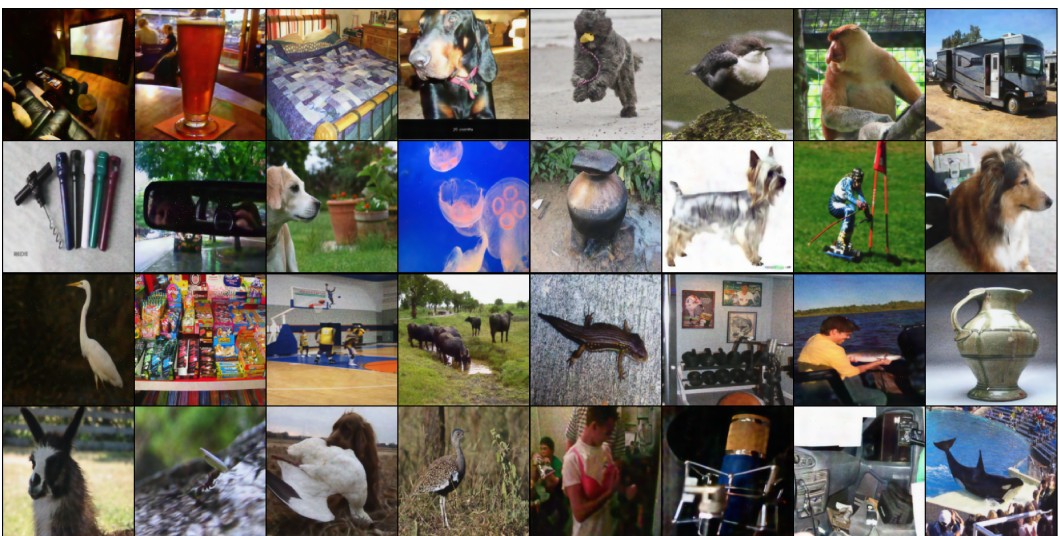

Figure 12: **Purified** ImageNet-1K images randomly selected for visualization

