# OpenReview forum: "MANI-Pure: Magnitude-Adaptive Noise Injection for Adversarial Purification"
_ICLR.cc/2026/Conference — ICLR 2026 Conference Withdrawn Submission_

### Official Review · Reviewer_YJi7 · 2025-10-15

**Soundness:** 3
**Presentation:** 2
**Contribution:** 1
**Rating:** 2
**Confidence:** 4

**Summary:**

Existing diffusion-based adversarial purification methods typically rely on uniform noise injection, which indiscriminately perturbs all frequencies, corrupting semantic structures and undermining robustness. This paper introduces MANI-Pure, a magnitude-adaptive purification framework that leverages the magnitude spectrum of inputs to guide the purification process. Extensive experiments on CIFAR-10 and ImageNet-1K validate the effectiveness of the proposed method.

**Strengths:**

1. The paper conducts extensive experiments across various attacks, metrics, and datasets.
2. The proposed method achieves state-of-the-art performance under most settings.

**Weaknesses:**

1. The paper has some readability issues: for example, the Abstract Section mentions “0.59%” and “2.15%” without specifying the corresponding datasets; FreqPure is incorrectly cited in line 83 (it should be Pei et al., 2025b [1]); and the sudden introduction of CLIP in line 95 may confuse readers.
2. Most of the technical components in the Methodology Section are based on existing work, showing limited novelty. In particular, compared with [1], the contribution appears incremental.
3. Although the authors claim that the proposed method incurs no training cost, DM-based adversarial purification methods typically introduce additional inference cost, which is not discussed. What is the inference cost of the proposed method?
4. As a DM-based adversarial purification method, the paper lacks evaluations against more diffusion-specific attacks or benchmarks [2,3].

[1] Pei et al. Diffusion-based adversarial purification from the perspective of the frequency domain. ICML 2025.
[2] Kang et al. DiffAttack: Evasion Attacks Against Diffusion-Based Adversarial Purification. NeurIPS 2023.
[3] Chen et al. Robust Classification via a Single Diffusion Model. ICML 2024.

**Questions:**

See Weaknesses.

---

### Official Review · Reviewer_eXCo · 2025-10-18

**Soundness:** 2
**Presentation:** 3
**Contribution:** 2
**Rating:** 4
**Confidence:** 5

**Summary:**

The paper introduces a novel diffusion-based defense framework that enhances adversarial robustness by adapting noise injection according to the input’s frequency characteristics. Unlike existing purification methods that apply uniform noise across all frequencies—often destroying semantic content—MANI-Pure analyzes the magnitude spectrum of adversarial images and injects heterogeneous, frequency-targeted noise to suppress perturbations concentrated in high-frequency, low-magnitude regions. It further integrates a complementary frequency purification module, which preserves low-frequency semantics while stabilizing high-frequency phase information during reverse diffusion.

**Strengths:**

1. This paper is well-written and well-structured.
2. The idea of injecting noise with different intensities in the frequency domain is relatively novel.
3. The paper provides a well-conducted and comprehensive literature review, and the compared baselines are all up-to-date and representative of the current state of the art.

**Weaknesses:**

1. MANI-Pure relies on a key observation — that adversarial perturbations are not uniformly distributed; they are predominantly concentrated in high-frequency regions, with heterogeneous magnitude intensity patterns that vary across frequencies and attack types. However, this motivation has already been explored in paper [1], which, in addition to studying the magnitude spectrum, also investigates the phase spectrum.
2. The method is unreasonable: after injecting noise of different intensities at different frequencies during the forward process, the reverse process does not remove this frequency-dependent noise. Instead, it employs an unrelated approach as the reverse process, which is conceptually inconsistent.
[1] Diffusion-based Adversarial Purification from the Perspective of the Frequency Domain

**Questions:**

No

---

### Official Review · Reviewer_svyq · 2025-10-22

**Soundness:** 3
**Presentation:** 3
**Contribution:** 2
**Rating:** 4
**Confidence:** 4

**Summary:**

**Summary**

The paper proposes an adaptive DBP framework, named **MANI-Pure**. Unlike prior DBP methods that apply uniform noise to adversarial examples, which has been studied as sub-optimal to purification process, the authors propose a magnitude-adaptive purification framework that purify adversarial perturbations based on a magnitude spectrum of the inputs. The experiment results show that the proposed framework improved the trade-off between clean accuracy and robust accuracy.

**Contribution**

1. The authors empirically show that adversarial perturbations are predominantly concentrated in high-frequency regions.

2. They propose MANI-Pure, a frequency-targeted purification method that suppresses high-frequency perturbations while preserving semantically meaningful low-frequency content.

Overall, the paper points out a promising direction of DBP methods to focus on frequency domain impact of adversarial perturbations. I hope the authors could address my concerns below to make the work more complete. I will raise my score if all my concerns are well addressed.

**Strengths:**

1. The paper is well structured, and the exploration of adversarial robustness from a frequency-domain perspective is both novel and promising.

2. The motivation is clearly articulated, where the introduction of MANI within the forward diffusion process is well justified.

3. The experimental evaluation is comprehensive, the proposed framework is thoroughly compared against recent DBP baselines, and an additional zero-shot experiment leveraging VLMs further strengthens the empirical evidence.

**Weaknesses:**

1. The attack setting is not comparable with prior works[1][2], see Q1, Q2 for details.

2. Prior works have shown empirical results of a predominant distributional discrepancy within attacked images on their amplitude and phase of high-frequency regions[3][4], where this weaken the empirical findings of Figure 1. Refer to Q3 below.

3. The explanation to the part (a) of Figure 2 could be expanded, see Q4, Q5 for details.


[1] "Diffusion Models for Adversarial Purification", ICML 2022.

[2] "Robust Classification via a Single Diffusion Model", ICML 2024.

[3] "Divide and conquer: Heterogeneous noise integration for diffusion-based adversarial purification", CVPR 2025.

[4] "Diffusion-based Adversarial Purification from the Perspective of the Frequency Domain", ICML 2025.

**Questions:**

1. In the main experiment, the iterative adaptive attacks (e.g., PGD+EOT) are only evaluated under a limited number of iterations. Prior work has shown that increasing the number of iterations is critical for assessing the robustness of newly proposed DBP methods under full gradient computations [1][2]. While computational resources may be constrained, I suggest evaluating at least $N = 100$ or $N = 200$ iterations for all adaptive attacks on the CIFAR-10 dataset.

2. Since DBP methods introduce additional randomness into the defense gradient [1], the robustness across varying numbers of EOT samples should be investigated as an ablation study to confirm stability. Appendix E.4 mentions following DiffPure’s settings, but recent works [1] have further discussed the EOT issue. Given that Figure 6 shows little variation across attack iterations, the authors should also clarify the gradient computation details in their framework.

3. Could the authors elaborate on the differences in empirical analysis between MANI-Pure and prior works (e.g., Freq-Pure [3])?

4. In Figure 1, the distribution of the magnitude spectrum appears skewed, suggesting significant disparity in $M_i$ values between early and later frequency bands. I suggest including a quantitative example of the band-wise weights $w_i$ to illustrate the distribution of magnitude importance.

5. The paper mentions that magnitude is averaged within each band. Does each band cover the same frequency range or contain an equal number of $(u, v)$ pairs? Since low-frequency regions tend to be more robust to adversarial perturbations, a more adaptive allocation of band-wise weights based on $M_i$ might improve performance. As Equation (9) appears somewhat heuristic (e.g., using an inverse correlation), I suggest discussing potential improvements in the weight assignment strategy.


[1] "Robust Classification via a Single Diffusion Model", ICML 2024.

[2] "Diffusion Models are Certifiably Robust Classifiers", NeurIPS 2024.

[3] "Diffusion-based Adversarial Purification from the Perspective of the Frequency Domain", ICML 2025.

---

### Official Review · Reviewer_K2uA · 2025-11-01

**Soundness:** 2
**Presentation:** 3
**Contribution:** 2
**Rating:** 4
**Confidence:** 4

**Summary:**

This paper observes that adversarial perturbations are not uniformly distributed. Motivated by this observation, this paper proposes MANI-Pure, a magnitude-adaptive DBP framework. In the forward process, MANI-Pure adaptively controls the noise injection based on the magnitude spectrum. In the reverse process, MANI-Pure decomposes low and high frequency components, aiming to preserve low-frequency content while focusing on purifying high-frequency content. Experiment results show that MANI-Pure achieves a better accuracy-robustness trade-off than baseline methods.

**Strengths:**

1. The paper identifies a key weakness of existing DBP methods: they inject uniform noise across all frequencies, which unnecessarily corrupts low-frequency content. The authors present a well-motivated solution by empirically showing that adversarial perturbations concentrate in high-frequency bands.
2. The proposed method achieves state-of-the-art robust accuracy among purification defenses. For example, on CIFAR-10 with a ViT-L/14 CLIP classifier, MANI-Pure outperforms prior DBP methods.
3. The experimental evaluation is thorough. The authors test across multiple datasets and multiple threat models, including strong adaptive attacks (e.g., PGD+EOT).

**Weaknesses:**

1. As mentioned in the related work, the observation that adversarial perturbations are not uniformly distributed is not new, and many studies have made efforts in addressing this issue. This makes the motivation of the paper weak.
2. Although the experiments include many baselines, it is notable that some very relevant recent methods from the literature are absent or only indirectly compared. For example, the authors mentioned [1] and [2] in the related work, but did not compare them with the proposed MANI-Pure in the experiments. It’s unclear if these approaches were unavailable, not easily integrable, or if the authors considered them similar enough to omit. Excluding them makes it harder to fully assess MANI-Pure’s novelty and advantage. If these methods offer comparable adaptive noise injection, the lack of direct comparison is a weakness
3. FreqPure is not the original idea of this paper, removing it will significantly decrease the performance. Meanwhile, the added benefit from MANI’s adaptive noising is relatively modest. The improvements from MANI can be seen as incremental tweaks rather than a breakthrough, especially considering that FreqPure is a well-established idea.
4. The paper does include adaptive attacks like BPDA+EOT and PGD+EOT, which is good, but it’s not explicitly shown how MANI-Pure handles an attack that violates its core assumption (e.g. an adversary intentionally injecting low-frequency distortions).
5. All classifiers tested are CLIP-based zero-shot models; it would be nice to see results on a standard trained classifier as well, to ensure the gains are not somehow specific to CLIP’s features. The focus on CLIP makes sense for plug-and-play usage, but evaluating on a typical model (e.g. a ResNet trained on CIFAR-10) could further demonstrate generality.
6. PGD+EOT is widely acknowledged as a stronger attack than BPDA+EOT for DBP methods [3], but the proposed method performs better on PGD+EOT than BPDA+EOT, which does not make sense to me. The iteration numbers for PGD, BPDA and EOT might be too low, resulting in ineffective attacks.

[1] Sample-specific noise injection for diffusion-based adversarial purification, ICML 2025.

[2] Diffcap: Diffusion-based cumulative adversarial purification for vision language models. arXiv preprint arXiv:2506.03933, 2025.

[3] Robust evaluation of diffusion-based adversarial purification. ICCV 2023.

**Questions:**

Please refer to the weaknesses.

---

### Note · Authors · 2025-11-12

I have read and agree with the venue's withdrawal policy on behalf of myself and my co-authors.